



# Competition for water vapour results in suppression of ice formation in mixed phase clouds

Emma L. Simpson, Paul J. Connolly, and Gordon B. McFiggans

Centre for Atmospheric Science, School of Earth and Environmental Sciences, University of Manchester, Manchester, M13 9PL, UK

*Correspondence to:* Emma Simpson (emma.simpson@manchester.ac.uk)

**Abstract.** The formation of ice in clouds can initiate precipitation and influence a cloud's reflectivity and lifetime, affecting climate to a highly uncertain degree. Nucleation of ice at elevated temperatures requires an ice nucleating particle (INP): so-called heterogeneous freezing. Previously reported measurements for the ability of a particle to nucleate ice have been made in the absence of other aerosol which will act as cloud condensation nuclei (CCN) and are ubiquitous in the atmosphere. Here

we show that CCN can 'outcompete' INPs for available water vapour thus suppressing ice formation, which has the potential to significantly affect the Earth's radiation budget. The magnitude of this suppression is shown to be dependent on the mass of condensed water required for freezing. Here we show that ice formation in a state-of-the-art cloud parcel model is strongly dependent on the criteria for heterogeneous freezing selected from those previously hypothesised. We have developed two alternative criteria which agree well with observations from cloud chamber experiments. This study highlights the dominant

role that competition for water vapour can play in ice formation, highlighting both a need for clarity in the requirements for heterogeneous freezing and for measurements under atmospherically appropriate conditions.

## 1 Introduction

A significant fraction, around 20% of clouds present in our atmosphere are mixed phase in that they contain both liquid and ice particles (Warren et al., 1986, 1988). Such clouds can be very persistent, (Morrison et al., 2012). As a result of their

widespread coverage, persistent nature and potential to greatly affect modeling results of cloud albedo and thus climate, (Sun and Shine, 1994), many observational and modelling studies have been dedicated to researching them, (eg. Verlinde et al. (2007); Hill et al. (2014); Korolev et al. (2003); Shupe et al. (2008)). Observations reveal processes within these clouds which we still do not fully understand. For example Morrison et al. (2012) describe the surprising persistence of Arctic mixed-phase clouds and the complex web of interactions between the many physical processes occurring in them that lead to their persistent

nature. Even with Morrison et al. (2012)'s insight into the nature of mixed-phase Arctic clouds the ability of climate models to simulate them is lacking, (Ovchinnikov et al., 2014). More research is required to fully understand the many ways aerosol composition and size distribution can influence mixed phase clouds, (de Boer et al., 2013). The number of ice crystals and liquid drops present in a cloud strongly depends on the size distribution of aerosols, (Twomey, 1991; Andreae and Rosenfeld, 2008). Aerosol particles may grow into liquid drops by the condensation of water vapour in water super-saturated environments. The



rate of growth and subsequent 'activation' into a cloud droplet of an aerosol particle is determined by its size and chemical composition (Köhler, 1936). There are two types of aerosol particles important for mixed-phase clouds; those that form cloud drops, cloud condensation nuclei (CCN), and those the can form ice crystals, ice nucleating particles (INPs). CCN are a subset of atmospheric aerosol particles and are ubiquitous in the atmosphere. They are generally made up of soluble compounds

allowing them to 'activate' into cloud droplets at relatively low super-saturations. INPs are much rarer and are required for freezing at temperatures above the homogeneous freezing level. Common INPs are mineral dusts (Murray et al., 2012) which are much less soluble making them less able to compete for water vapour than CCN, as an INP of the same size as a CCN would 'activate' into a cloud droplet at a higher super-saturation. This difference in ability to compete for water vapour is potentially significant as it is thought that in mixed-phase clouds the most effective INPs are contained within a liquid drop,

(Field et al., 2012; Murray et al., 2012).

In this study it is hypothesised, and demonstrated with a cloud parcel model, that the presence of CCN within a cloud could suppress the formation of ice. The ability of CCN to 'outcompete' INPs for water vapour means they could activate into cloud droplets, and therefore provide a sink for water vapour, before the maximum supersaturation within the cloud has reached that that would allow INPs to grow to an appreciable size. This would result in the INPs without sufficient water on them to be able

to freeze, assuming liquid water is a requirement for freezing of the most effective INPs.

In this study we demonstrate that the presence of CCN can reduce the ice nucleating potential of INPs as a result of the competition for water vapour. Although the idea that competition for water vapour can result in a reduction of ice formation has been discussed in previous studies, here we demonstrate that it plays an important role in cloud parcel model simulations of mixed-phase clouds where the presence of atmospherically relevant CCN concentrations can significantly reduce ice crystal

number concentrations. Levin et al. (2016) observed that competition between aerosol particles limited the targeted RH within a Continuous Flow Diffusion Chamber (CFDC) resulting in inaccurate measurements of INP concentrations. Here we show that the competition for water vapour is not only important in instruments measuring INP number concentrations but also in expansion chambers which are more atmospherically representative than CFDCs as well as in simulations using a detailed parcel model. Previous experimental results from the AIDA expansion chamber which have been used to measure the ice

nucleating ability of several different particle types have corrected for the fact that not all aerosol in chamber will have been able to activate and grow into cloud droplets due to competition for water vapour, (Ullrich et al., 2017). This means that their formulation of INP parameterisations do not take competition for water vapour into account as it has already been corrected for. Therefore the competition effect needs to be addressed within the models which use these ice nucleation parameterisations. Modelling studies such as that by de Boer et al. (2013) do consider the concentration of CCN as having an influence on

ice formation in clouds and show that higher concentrations of CCN can reduce ice number concentration. However their explanation of this effect differs from the one put forward here. de Boer et al. (2013) explain that the reduction in ice formation in their simulations preformed with high CCN concentrations is due to a reduction in drop volume, similar to the Twomeny effect, (Twomey, 1974). The immersion freezing parameterisation that de Boer employ is a function of drop volume and the production of ice particles through immersion freezing decreases as drop volume decreases. In this study we use a different



method for calculating the freezing rate that follows Connolly et al. (2009) and Niemand et al. (2012) and is based on the dry surface area of an INP not drop volume.

Ice nucleation initiated by an INP can take place via either deposition nucleation or freezing nucleation, (Vali et al., 2015). Freezing nucleation requires the presence of liquid water on an INP, and it has also been argued that deposition nucleation

has a liquid water transition phase, (Vali et al., 2015) however the exact amount of liquid that needs to be present for ice to form remains unconstrained. In the literature, immersion freezing (which is a type or 'mode' of freezing nucleation) has been confined to those particles that have activated in cloud drops, (Hoose et al., 2010a; Pruppacher and Klett, 1997) or those drops greater than $2\mu$m in diameter, (Paukert and Hoose, 2014). Diehl and Wurzler (2004) use a water activity, similar to the Koop et al. (2000) approach to calculate amount of condensed water required to overcome the freezing point depression caused by

soluble compounds present as an internal mixture within the INP. However de Boer et al. (2013) found that the freezing point depression due to soluble compounds was not important for large cloud droplets such as those found in stratiform clouds, since droplets are large enough to overcome the effect.

We conducted experimental and modelling studies aiming to investigate the competition between INPs and CCN for water vapour under conditions suitable for mixed-phase clouds. A comparison of the different criteria for immersion freezing is also

made. A description of the methodology is provided in the modelling Sect. 2 and the experimental Sect. 3. Section 4 provides a demonstration of the suppression effect using parcel model results. Results from a sensitivity study on the conditions where the suppression effect is greatest are given in Sect. 5 followed by a summary of the overall findings of this work in Sect. 6.

## 2 Model Description

ACPIM is a detailed bin-resolving cloud parcel model which can be used to model particle activation, droplet growth and ice

nucleation within a rising parcel of air under going adiabatic ascent. It can also be used to model particle activation during an expansion experiment in which cloud conditions are generated within a cloud chamber such as the Manchester Ice Cloud Chamber, (MICC). When modelling a rising parcel of air, water vapour is made available for condensation by prescribing a constant updraft velocity and assuming an atmosphere in hydrostatic balance. For modelling expansions in a chamber, pressure and temperature drop rates provide the source of super-saturation. When modelling specific experiments in a cloud chamber,

time dependent pressure and temperature drop rates can be prescribed so that the simulated temperature and pressure profiles fit those that were observed (see Fig. 1. for an example). An initial relative humidity is defined in the model for both types of simulations.

ACPIM allows the size distribution of any composition of aerosol to be defined using lognormal size distributions. A log-normal size distribution describes the number of aerosol particles per natural logarithm of the bin width, $\frac{dN}{dlnD_p}$, as is given by

the following equation,

$$\frac{dN}{d\ln D_p} = \frac{N_{ap}}{\ln\sigma\sqrt{2\pi}}\exp\left[-\frac{\ln^2\left(\frac{D_p}{d_m}\right)}{2\ln\sigma^2}\right] \tag{1}$$



where $N_{ap}$ is the total number concentration of aerosol particles, $\ln \sigma$ is the natural logarithm of the geometric standard deviation and $d_m$ is the median diameter (Jacobson, 1999). Any number of aerosol size distributions, or 'modes', can be included. Each mode can be made up of any composition. This allows for both internally and externally mixed aerosol size distributions to be represented in the model.

Both sub- and super-saturated growth of aerosol particles is included in the model. The point at which an aerosol particle 'activates' into a cloud droplet depends on its size and chemical composition and is traditionally defined by Köhler Theory (Köhler, 1936; Pruppacher and Klett, 1997). Köhler Theory is used to calculate the 'critical supersaturation' and 'critical droplet radius' required for a given aerosol particle to activate into a cloud droplet. Köhler Theory is often approximated by $\kappa$-Köhler Theory (Petters and Kreidenweis, 2007) which offers a simplified approached for use in modelling cloud-aerosol

interactions. Sorjamaa and Laaksonen (2007) and Kumar et al. (2009) present an activation theory based on the multilayer adsorption of gases which shows promise for the treatment of insoluble particles, (Kumar et al., 2011). In all theories the critical supersaturation with respect to water and the critical wet diameter of particle required for activation is dependent on its size and composition. The saturation ratio of water in equilibrium with a particle, for all particles that include some soluble mass in this study, is defined using $\kappa$-Köhler Theory following Petters and Kreidenweis (2007), Eq. (2) below. This equation

is used to calculate a particle's critical supersaturation with respect to water and its critical wet diameter.

$$s = a_w exp\Big(\frac{4\sigma_{\frac{s}{a}} M_w}{RT\rho_w D}\Big) \qquad (2)$$

where,

$$a_w = \frac{D^3 - D_p^3}{D^3 - D_p^3(1-\kappa)} \qquad (3)$$

$s$ is the saturation ratio of a particle with dry diameter $D_p$, $\sigma_{\frac{s}{a}}$ is the surface tension of the solution/air interface = 0.072

$Jm^{-2}$, $M_w$ is the molecular weight of water, $R$ is the universal gas constant, $T$ is temperature, $\rho_w$ is the density of water and $D$ is the particle's wet diameter. Values for the constant $\kappa$ are obtained from measurements for different particle types. Here a value of 0.61 is used for ammonium sulphate (Petters and Kreidenweis, 2007). Insoluble species are given a $\kappa$ value of zero, (Petters and Kreidenweis, 2007). The overall $\kappa$ value for an internally mixed aerosol particle is calculated according to the simple mixing rule, Eq. (4), (Petters and Kreidenweis, 2007).

$$\kappa = \sum_i \varepsilon_i \kappa_i \qquad (4)$$

Where $\varepsilon_i$ is the volume fraction of component $i$ in the particle. The growth of drops follows Pruppacher and Klett (1997) and Jacobson (1999) and includes kinetic limitations to growth important for large aerosol particle sizes, (Simpson et al., 2014). Freezing rates of INPs is governed by the $n_s$ parameterisation following Connolly et al. (2009) and Niemand et al. (2012),





where $n_s$ is the number concentration of ice active sites per unit surface area of an INP. Here $n_s(T)$ takes the following form for K-feldspar

$$log_{10}(n_s(T)) = -aT + b \tag{5}$$

where $a = -0.1963$, $b = 60.2118$ and $T$ is temperature in degrees Kelvin. The values for $a$ and $b$ were measured for K-feldspar by Emersic et al. (2015), from the same sample as used in chamber experiments in this study, in the MICC. $n_s(T)$ takes the following form for desert dust

$$n_s(T) = exp[a(T - 273.15) + b] \tag{6}$$

where $a = -0.517$ and $b = 8.934$ and $T$ is temperature in degrees Kelvin. The values for $a$ and $b$ are from Niemand et al. (2012) and were measured for a variety of natural dust samples. This parameterisation for desert dusts is valid for use in the temperature range of -12 to -36$^o C$, (Niemand et al., 2012).

Before an aerosol particle can nucleate ice (in all modes except deposition (although it has been suggested that there exists a liquid transition phase in deposition freezing (Vali et al., 2015))) it must be in contact with some liquid water. The minimum mass of water required for ice formation is currently unconstrained. Here we define a 'heterogeneous freezing criteria' in order to prevent aerosol particles without any condensed water on them from freezing in the model. In this study several heterogeneous freezing criteria are compared, detailed in Table 1. For all criteria once the criteria has been achieved, freezing is determined by the value of $n_s(T)$ and the surface area of the particle.

| Label | Description | References |
|---|---|---|
| Activated only | Only activated drops can freeze | Hoose et al. (2010a) Pruppacher and Klett (1997) |
| $M_{cw}$ | Drops must have a threshold mass of water which is dependent on particle size | (*this study*) |
| $A_w$ | Drops must have a threshold water activity, $A_w$ | (*this study*) |
| RH >1 | At RH < 1 drops cannot freeze | Field et al. (2012) |

**Table 1.** Table describing the different criteria for heterogeneous freezing used in this study. Once the criteria has been achieved, freezing is determined by the value of $n_s(T)$ and the surface area of the particle.

There are many contrasting definitions and treatments of the different pathways, or *modes*, of heterogeneous freezing found in the literature. Four distinct modes are described: immersion, condensation, contact and deposition. Physical similarities between the different modes can make it difficult for distinctions to be made when modelling heterogeneous freezing. For example contact, immersion and condensation all require liquid water to be present. It is common for two modes of freezing to be treated together in modelling studies. For example Hoose et al. (2008) treat immersion and condensation freezing together,





whereas Morrison et al. (2005) argue that condensation and deposition freezing are similar so treat them as the same mode of freezing. Wex et al. (2014) also discuss the lack of clear definitions of the different modes of freezing in the literature. There is uncertainty in the definition of the requirements for immersion or condensation freezing. Walko et al. (1995) and Kärcher and Lohmann (2003) state that condensation freezing requires water saturation; however, Ervens and Feingold (2012) describe

condensation freezing to occur below water saturation and immersion to happen above. To avoid confusion and to keep a physical as possible representation for freezing we do not make a distinction between the different modes of freezing. Instead by defining a 'freezing criteria' (see Table 1) insures that an INP has an amount of water condensed onto it before it is able to freeze. Now follows a description of the freezing criteria compared in this study.

*Activated only* − Activation marks the point at which an aerosol particle grows rapidly by the condensation of water vapour,

and is often referred to as the point at which an aerosol particle becomes a cloud droplet. Using activation as a criteria for heterogeneous freezing insures that an INP is immersed within a liquid drop and therefore can only act in the immersion/condensation freezing modes. However the mass of water required for activation is not related to ice formation and INPs which are below activation size may still have sufficient water on them to be able to freeze as the critical mass of water required for freezing is currently unconstrained.

*RH > 1* − this criteria for freezing insures that almost all aerosol types will have some condensed water on their surface and therefore insuring INPs only nucleate ice in the immersion/condensed modes. This criteria for freezing does not provide a specific mass of liquid water required for freezing meaning that an INP may only have a very small mass of water condensed onto it, depending on the particle's hygroscopicity, before it initiates freezing. This criteria also does not take into account the effects of solutes on the freezing of a solution drop.

$M_{cw}$ − In this study we have developed two new criteria for freezing which are both based on the idea that a threshold mass of water is required for freezing. The first of these new criteria defines a threshold mass which is only dependent on particle dry size and is calculated according to,

$$M_{cw} = \frac{\alpha_{cw}}{6} \pi D_p^3 \rho_w \qquad (7)$$

where $M_{cw}$ is the critical water mass required for freezing, $\alpha_{cw}$ is a constant, $D_p$ is the particle dry diameter and $\rho_w$ is the

density of water. The constant $\alpha_{cw}$ in this study has a value of 70. This value gave best agreement between model results for ice number concentration and chamber measurements across 6 experiments (Fig. 3.). Values for $\alpha_{cw}$ below 20 and above 125 gave poor agreement with chamber results. A size dependent threshold mass of condensed water insures that the liquid layer on the surface of an INP is a minimum depth before ice nucleation can take place. This criteria prevents an INP without any water on it from freezing. The physical idea behind this criteria for freezing is that ice nucleation requires the formation of a

ice-like cluster of water molecules to reach a critical size to initiate freezing, (Fitzner et al., 2015). This criteria is used to insure some water mass is present on an INP to allow for the formation of a critical cluster. However the value of $\alpha_{cw}$ (used in the calculation of $M_{cw}$) found in this study to give best agreement with observations suggests a threshold mass of water required



for freezing much larger than that used by Fitzner et al. (2015) modelling studying on ice nucleation. Further investigation is required in order to establish an exact value for the threshold mass of water required for freezing.

$A_w-$ The second new criteria for freezing is a threshold water activity, $A_w$. Water activity is calculated according to Eq. (3) and is related to wet size and composition of a particle. The value for the threshold $A_w$ which gave best agreement between
model and chamber results across 6 experiments, (Fig. 3.) was found to be 0.9999. Values for the threshold $A_w$ below 0.9997 and above 0.99994 did not agree with chamber results. A threshold water activity required for freezing effectively defines a threshold water concentration required for freezing. This criteria insures an INP is immersed in a liquid drop and sets a threshold for the amount of solute presented in the drop. This criteria explicitly takes the effects of solutes on freezing into account. Koop et al. (2000) also use water activity to determine homogeneous freezing rate.
These two new criteria for freezing are potentially more physically related to ice formation than the other two criteria described in Table 1 as they are associated with the ratio of particle size to condensed water mass. It is physically reasonable to suggest that a specific ratio of water mass to particle mass is required for freezing. In order to make the new criteria generally applicable to all INP types, further experiments are required to constrain the values of $\alpha_{cw}$ and threshold $A_w$. This work provides a 'proof of concept' for the new criteria and a demonstration of the impact heterogeneous freezing criteria can have
on the number concentration of ice crystals predicted by cloud microphysical models.

In this study, contact nucleation is not considered, as collisions between particles do not take place in the model set-up used here. Due to the short duration of chamber experiments carried out here as well as the relatively small droplet sizes, collisions between particles in the chamber are unlikely to occur, (Rogers and Yau, 1989). Contact nucleation is also not considered in adiabatic parcel simulations as contact nucleation is thought to be of secondary importance compared to the other modes of
freezing, (Phillips et al., 2007).

The full moving bin structure (Jacobson, 1999) was used within the model. This structure was chosen as it is the least numerically diffusive, (Jacobson, 1999). It was found that under certain conditions the number concentration of ice crystals was sensitive to the bin structure used, see Supplementary Fig. S1. Currently in ACPIM the full moving bin structure does not allow interactions between particles to take place. Here we are only interested in the condensation of water vapour onto
particles and the subsequent nucleation and growth of cloud drops and ice crystals.

## 3   Chamber Experiments

### 3.1   Description of the chamber and experiments

MICC is a 10m tall and 1m in diameter steel tube housed within the University of Manchester. It spans three floors and on each floor is contained within a cold room where the temperature can be controlled between approximately 20$^o$C and
-50$^o$C. Air can be evacuated from the chamber using two scroll pumps. This evacuation of air causes a cooling within the chamber which generates cloud conditions. The experiments conducted within the current study used the following procedure: 1) Several cleaning cycles were preformed in the chamber until the particle concentration in the chamber was below 10 cm$^{-3}$, typically around 5 cm$^{-3}$. 2) INP, in this case K-feldspar, was introduced to the chamber via a dust generator at the top of





the chamber. Total aerosol number concentrations were measured at the bottom of the chamber using a Condensation Particle Counter (CPC). A measurement of the aerosol size distribution using a Scanning Mobility Particle Sizer 3081 TSI and Grimm 1.109 was also made at this stage, both sampling at the bottom of the chamber. When the concentration measured by the CPC had stabilised the aerosol in the chamber was assumed to be well mixed. 3) Temperature within the chamber was measured by 8 thermocouples at locations long it's entire length. When all thermocouples measured the same temperature, +/- 0.5$^o$C, the chamber was ready for an expansion experiment to begin. 4) The pumps located at the top of the chamber were switched on, to evacuate the chamber, as was a Droplet Measurement Technologies Cloud Droplet Probe (CDP) and a Stratton Park Engineering Company, Inc., Cloud Particle Imager (CPI) Version 1 instrument sampling at the bottom of the chamber. Air was evacuated down to 700 mb within the chamber. 5) Air containing ammonium sulphate aerosol was then added to the chamber via a pipe connected to the Manchester Aerosol Chamber (MAC). The pressure in the MICC was then returned to atmospheric pressure by filling from MAC or the clean air system. 6) Steps 3 and 4 of the process were then repeated with the mixture of ammonium sulphate and K-feldspar aerosol.

### 3.2 Model set-up and initial conditions for chamber experiments

For each expansion the temperature profile in the model was fit to the lowest temperature measurements made in the chamber during an expansion, an example is given in Fig. 1b. It was chosen to fit to the lowest temperature measurements in the chamber as this is likely where most ice would form. An example of the pressure profile in the chamber and the model is given in Fig. 1a. Also shown in Fig. 1. are the corresponding temperature and pressure profiles from model simulations of those expansion experiments. Both profiles in Fig. 1. as well as the agreement between measurements and model fit are typical of all expansion experiments presented in Fig. 3.

The initial RH in the chamber was calculated so that water saturation was reached in the model at the same time as was reached in the chamber. Water saturation was assumed to be reached in the chamber when the concentration of drops (particles > 5 $\mu$m measured by the CDP) rapidly started to increase, indicating droplet activation.

Lognormal size distributions where fitted to aerosol measurements (examples are given in Supplementary Fig. S3 and Fig. S4.) and used as input to ACPIM for each chamber expansion experiment.

### 3.3 Classification of particle type in chamber simulations

A CPI was used to make a qualitative assessment of the phase, liquid or ice, of hydrometeors formed during chamber expansions. Due to the small size and high number concentration of the hydrometeors formed in experiments, data from the CPI could not be used quantitatively because of the instrument's small sample volume.

The peaks in concentration of particles with sizes greater than 12 $\mu$m as measured by the CDP are due to the formation of drops at the beginning of expansions. The CPI measures spherical particles, drops, early in expansions, before measuring small irregularly shaped particles, ice, (see Fig. 2a.). The drops formed at the beginning of expansions quickly evaporate, as ice crystals begin to form, due to the Wegener-Bergeron-Findeisen (WBF) process. This explains the peak in concentration




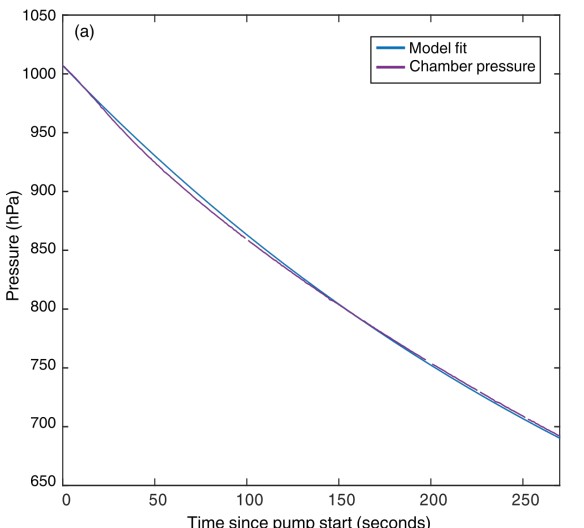
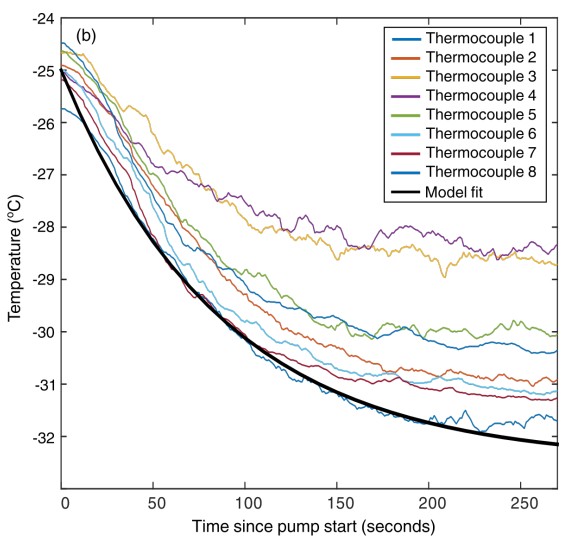

**Figure 1.** Example fits for pressure and temperature during an expansion experiment. a) The purple line is the pressure measured in the chamber by a Keller pressure sensor. The blue line is the pressure as simulated by ACPIM. b) Coloured lines show the temperature inside the chamber measured by 8 different thermocouples spread throughout the height of the chamber. The black line is the temperature as simulated by ACPIM.

towards the beginning of expansions, Fig. 3. and also explains why the modelled ice number should be compared with measured particles later in the simulations, since WBF process ensures that ice and liquid water will not coexist.

A size threshold of 12 $\mu$m was chosen as the distinction between liquid and ice particles. A clear distinction between supercooled drops and ice crystals can be seen in Fig. 2b. Similar experiments carried out in MICC with the same set-up and K-feldspar sample were carried out by Emersic et al. (2015) they too chose a value close to 12 $\mu$m to distinguish between ice and liquid.

### 3.4 Chamber results

A total of 6 expansion experiments were carried out. Where CPI data was available dashed blue lines represent the sum of liquid droplets and ice crystals initially formed. In experiments where CPI data is not available (Fig. 3. b,d,f) it can be assumed that drops evaporate at a similar expansion time (around 80 seconds) as in experiments where CPI data is available (Fig. 3. panels a, c and e), because the experimental conditions are the same. The aerosol initial conditions for each expansion experiment in Fig. 3. are given in Table 2. Ammonium sulphate aerosol was included in some of the expansions (Fig. 3. panels c through f) with the aim of observing the suppression effect. However no suppression of ice was observed. This is because in this area of the parameter space, particle sizes around 0.4 $\mu$m and moderate updraft velocities similar to the pressure drop rates in the chamber, little suppression is seen as found in model simulations when using either of the three criteria for freezing compared in Sect. 5.





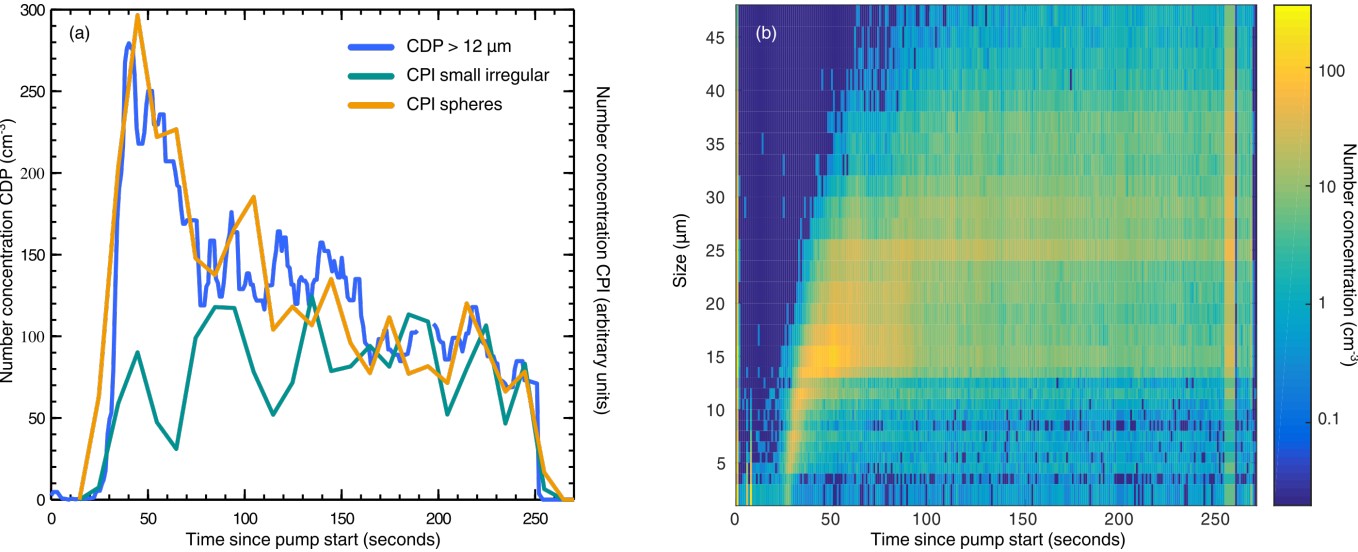

**Figure 2.** a) Measurements from the CPI instrument for the two particle habit classifications observed, small irregular (ice), green line, and spheres (drops), orange line, and the CDP for particles with sizes greater than 12 $\mu$m, blue line. CPI data is shown without units because the data is qualitative. b) Size distribution of cloud particles measured by the CDP in MICC during an expansion experiment.

Figure 3 panels a and b show the number concentration of ice crystals, as measured by the CDP, in the cloud chamber during expansions with only K-feldspar aerosol. Due to the low temperature of the experiments, -25$^o$C, many ice crystals are formed resulting in small ice particles and few drops. As described in Sect. 2 collision coalescence is not thought to play a significant role in the growth of droplets in the chamber experiments described here therefore it is unlikely that there will be many drops

in the chamber larger than around 12$\mu$m. In order to distinguish between liquid and ice particles measured by the CDP, it is assumed all particles above 12 $\mu$m in diameter are ice, see Fig. 2b. Results from four ACPIM simulations, with the same initial conditions as the chamber expansion, each using a different criteria for heterogeneous freezing, detailed in Table 1, are also shown in Fig. 3.

Figure 3 demonstrates that the two new criteria for freezing, $M_{cw}$ and $A_w$, agree most consistently with observations out

of the four criteria compared. $RH > 1$ tends to predict higher ice crystal number concentrations than the other criteria, and over estimates the number of ice crystals in the chamber by a factor of around 2 in panels a and b. *Activated only* criteria also over estimates the number of ice crystals in panels a and b however significantly under estimates the ice crystal number concentration in panels e and f. This under estimation is due to the initial onset of ice formation in model when using the *Activated Only* criteria. Ice formation occurs before most aerosol have activated, which results in too few ice crystals forming

as the WBF processes causes drops to evaporate before they can activate.





**Figure 3.** Number concentration of ice crystals in 6 chamber experiments and model simulations using different criteria for freezing. The solid blue line is the number concentration of ice measured in the chamber, dashed blue line represents the sum of liquid droplets and ice crystals initially formed and the other coloured lines are simulated ice number concentrations using different criteria for freezing in the model.





| Fig. Panel | K-feldspar | | | Ammonium Sulphate | | |
|:---:|:---:|:---:|:---:|:---:|:---:|:---:|
| | $N$ (cm$^{-3}$) | $D$ ($\mu$m) | $\ln \sigma$ | $N$ (cm$^{-3}$) | $D$ ($\mu$m) | $\ln \sigma$ |
| a | [680, 17] | [0.35, 1.5] | [0.42, 0.4] | 0 | 0 | 0 |
| b | [900, 55] | [0.35, 1.5] | [0.45, 0.4] | 0 | 0 | 0 |
| c | [330, 12] | [0.35, 1.5] | [0.42, 0.4] | 6200 | 0.07 | 0.4 |
| d | [720, 35] | [0.35, 1.5] | [0.48, 0.4] | 6000 | 0.07 | 0.5 |
| e | [240, 7] | [0.35, 1.5] | [0.4, 0.4] | 9200 | 0.07 | 0.43 |
| f | [400, 17] | [0.35, 1.5] | [0.43, 0.4] | 6500 | 0.07 | 0.5 |

**Table 2.** Summary of aerosol initial conditions for experiments in Figure 3. The two values listed between the square brackets are the parameters for the two lognormal modes of aerosol present.

## 4 Demonstration of the Suppression Effect

To demonstrate the suppression of ice formation by the presence of CCN, two simulations were preformed with ACPIM: one with INPs and a small number of CCN and the other with the same number of INPs and a higher number concentration of CCN. The two cases are referred to as *low CCN* and *high CCN*, respectively. Results for *low CCN* and *high CCN* cases where the criteria for freezing is *activated only*, are shown in Fig. 4a and 4b. The *low CCN* case contained 1 L$^{-1}$ 300 nm K-feldspar and 50 cm$^{-3}$ 60 nm ammonium sulphate particles, K-feldspar is the source of INPs and ammonium sulphate particle represent CCN. The *high CCN* case contained the same K-feldspar aerosol and 2000 cm$^{-3}$ 200 nm ammonium sulphate. The inclusion of some CCN particles in the *low CCN* case was necessary in order to be atmospherically relevant, atmospheric aerosol concentrations as low as 1 L$^{-1}$ are not found and 1 L$^{-1}$ is considered a typical atmospheric concentration of INPs, (Murray et al., 2012). Although such low CCN concentrations as 50 cm$^{-3}$ are still unrealistically low for most of the atmosphere, with the exception of the Arctic ocean, (Mauritsen et al., 2011), it is a situation that maximises the chance for ice formation and at which no suppression is found. The initial conditions for the simulations are the same as those in Table 3 with an updraft velocity of 0.5 ms$^{-1}$.

Figure 4a shows a reduced ice number concentration in the *high CCN* case, orange line. This is due to the CCN 'outcompeting' the INPs, and providing a sink for water vapour, thus reducing the maximum supersaturation in the cloud below the critical supersaturation of the INPs. The reduction in relative humidity in the *high CCN* case can be seen in Fig. 4b. The liquid water mixing ratio in both cases are similar (see Supplementary Fig. S2) showing that the same amount of water is condensing in both cases but onto different particles.

The same *high CCN* and *low CCN* cases, with the same initial conditions, were run with different criteria for heterogeneous freezing. Assuming freezing can only occur above water saturation, as in previous studies by de Boer et al. (2010) and as observed by Ansmann et al. (2008), shows no suppression of ice formation. The number concentration of ice in the *high CCN*





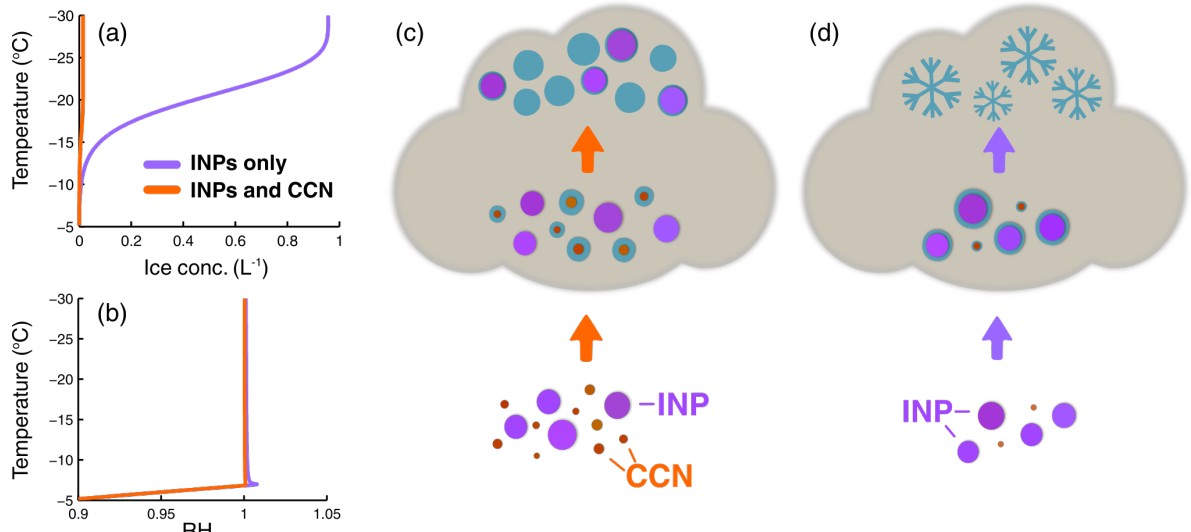

**Figure 4.** A schematic diagram demonstrating the suppression of ice formation caused by the presence of CCN. c) shows the INPs and CCN (*high CCN*) case where water vapour condenses onto the CCN in preference to the INPs preventing them from activating and freezing. d) shows the low CCN case where, without the CCN, the max super-saturation (a) purple line) can rise high enough for INPs to activate and then freeze. a) show relative humidity and b) ice concentration with decreasing temperature for the two cases. The *low CCN* case (purple lines) contained 1 L$^{-1}$ 300 nm K-feldspar and 50 cm$^{-3}$ 60 nm ammonium sulphate particles. The *high CCN* case (orange lines) contained 1 L$^{-1}$ 300 nm K-feldspar and 2000 cm$^{-3}$ 200 nm ammonium sulphate. Initial conditions for the two simulations are the same as those detailed in Table 3. An updraft velocity of 0.5 ms$^{-1}$ was used in both cases. The criteria for heterogeneous freezing *activated only*.

and *low CCN* cases is the same. This assumption does not depend on the amount of water condensed onto an INP, therefore the ability of an INP to compete for water vapour is irrelevant for ice formation.

## 5 Sensitivities of the Suppression Effect

An investigation to find the conditions under which the suppression effect is most significant was conducted. The number

5 concentration of ice crystals in several pairs of simulations of a rising parcel of air, with initial conditions detailed in Tables 3 and 4, one with INPs and a small number of CCN present, *low CCN*, the other with more CCN and the same number of INPs, *high CCN* simulations, were compared. A small amount of CCN where included in the *low CCN* cases as total atmospheric aerosol concentrations as low as 0.001 cm$^{-3}$ do not occur. Ice number concentrations where taken at the simulation time that corresponded to a temperature -30$^o$C for results shown in Fig. 5. The results from these simulations of the percentage difference

10 in ice number concentration between the *low CCN* simulations and *high CCN* simulations are shown in Fig. 5 for INPs with four different soluble fractions:- 0%, 1%, 25% and 50%.





| Parameter | Value |
|-----------|-------|
| Pressure | 800 hPa |
| Temperature | -5$^o$C |
| RH | 90% |

**Table 3.** Initial conditions for parcel model simulations

For zero soluble fraction $\kappa$-köhler theory is replaced by FHH adsorption theory following Kumar et al. (2009). FHH adsorption theory may be a more appropriate approach to treating completely insoluble particles, (Kumar et al., 2011). FHH theory is similar to $\kappa$-Köhler theory; however, instead of using a single constant $\kappa$ to represent the hygroscopicity of a particle two factors, A$_{FHH}$ and B$_{FHH}$, are used to represent molecules adsorbing onto the surface of an insoluble aerosol particle. A$_{FHH}$

represents the interactive forces of water molecules between the surface and adjacent adsorbate molecules, (Hatch et al., 2014). B$_{FHH}$ characterises the attractive forces between the surface and subsequent adsorbed water layers, (Hatch et al., 2014). In ACPIM for an aerosol particle with a soluble fraction of zero, the equation for its equilibrium saturation ratio is calculated using the following equation,

$$s = exp\left(\frac{4\sigma M_w}{RT\rho_w D}\right) exp(-A_{FHH}\Theta^{-B_{FHH}}) \tag{8}$$

Where $\sigma$ is the surface tension at the particle-gas interface, $M_w$ is the molecular mass of water, $R$ is the universal gas constant, $T$ is temperature, $\rho_w$ is the density of water and $D_p$ is the particle diameter. $\Theta$ is the number of adsorbed layers, defined as the number of adsorbed water molecules divided by the number of molecules in a monolayer, Kumar et al. (2009). Values for A$_{FHH}$ and B$_{FHH}$ are determined experimentally and are unique to different compounds, (Kumar et al., 2009). The values for the A$_{FHH}$ and B$_{FHH}$ constants used here are 2.25 and 1.8 respectively and are from measurements made by Kumar

et al. (2011) on a variety of dust samples.

Figure 5 highlights the locations in the parameter space where most suppression occurs when the criteria for heterogeneous freezing is *Activated Only*. A value of 100 % suppression indicates no ice formation in the *high CCN* case compared to between 0.03 and 0.7 L$^{-1}$ ice crystal number concentration in the *low CCN* case. Figure 5a. shows results for the percentage suppression of ice due to the presence of CCN when the INPs are completely insoluble and their growth and activation into cloud drops is

calculated using FHH adsorption theory. Most suppression occurs at low updraft velocities and small median diameters of INP.

Figure 5b. shows similar results to Fig. 5a. however for INPs that have a 1 % soluble fraction. In this case the growth and activation into cloud drops is calculated according to $\kappa$-köhler theory. The soluble fraction is made up of ammonium sulphate. There is a general trend towards less suppression in panel b compared to panel a. This indicates that the slightly soluble INPs are better able to compete for water vapour. In the top right hand corner, high updraft and large INP diameters, of panel b there





| | | high CCN | low CCN |
|---|---|---|---|
| INP | $N$ (#$cm^{-3}$) | 0.001 | 0.001 |
| | $D$ (nm) | (variable) | (variable) |
| | $ln\sigma_g$ | 0.5 | 0.5 |
| Mode 2 | $N$ (#$cm^{-3}$) | 185 | 60 |
| | $D$ (nm) | 26 | 60 |
| | $ln\sigma_g$ | 0.44 | 0.45 |
| Mode 3 | $N$ (#$cm^{-3}$) | 1364 | - |
| | $D$ (nm) | 85 | - |
| | $ln\sigma_g$ | 0.47 | - |
| Mode 4 | $N$ (#$cm^{-3}$) | 276 | - |
| | $D$ (nm) | 246 | - |
| | $ln\sigma_g$ | 0.32 | - |

**Table 4.** Log-normal parameters for the aerosol in both *low CCN* and *high CCN* simulations. The *high CCN* aerosol are from Van Dingenen et al. (2004) from measurements taken during summer months in the afternoon with only *natural* sources. Mode 1 in *low CCN* is made up of K-feldspar aerosol, (with varying soluble fraction made of ammonium sulphate) and Mode 2 is ammonium sulphate.

is a slight suppression. This feature becomes more prominent at higher soluble fractions as can be seen in panels c and d of Fig. 5.

Most suppression occurs at low updraft velocities and small INP diameters. This is to be expected because smaller diameter particles require higher super-saturations in order to activate into cloud droplets and then freeze. At low updraft velocities the maximum supersaturation generated in the cloud will be low so that only particles with low critical super-saturations, i.e. hygroscopic, such as those of CCN, can activate. The activation of CCN into cloud drops creates a sink for water vapour keeping the maximum super-saturation in the cloud below that of the INPs thus preventing them for activating and freezing.

Figure 5 reveals two regimes that result in the suppression of ice formation due to the presence of CCN. The first regime, Regime 1, is when INPs are in competition with CCN for available water vapour, bottom left of panels in Fig. 5. The CCN are better able to compete than the INPs and thus grow and activate into cloud drops creating a sink for water vapour and thus prevent the maximum supersaturation in the parcel rising to that which would allow the INPs to activate. As only activated drops can freeze in this case, ice formation is suppressed since INPs are prevented from activating. In the second regime, Regime 2, towards the top right hand corners of panels c and d in Fig. 5., INPs have a significant soluble fraction that allows them to act as giant CCN at large diameters, towards 2 $\mu$m. This results in a reduction of ice formation in *high CCN* simulations and, to a lesser extent, in *low CCN* simulations, which leads to a suppression of ice formation because the INPs are in competition with themselves as well as the CCN for available water vapour. At higher updraft velocities this suppression is enhanced because less INPs can activate due to the kinetic limitations to growth of large aerosol particles and the reduced simulation time required to reach -30$^o$C at higher updraft velocities. The ice crystal number concentration in *low CCN* cases at high updrafts and large sizes





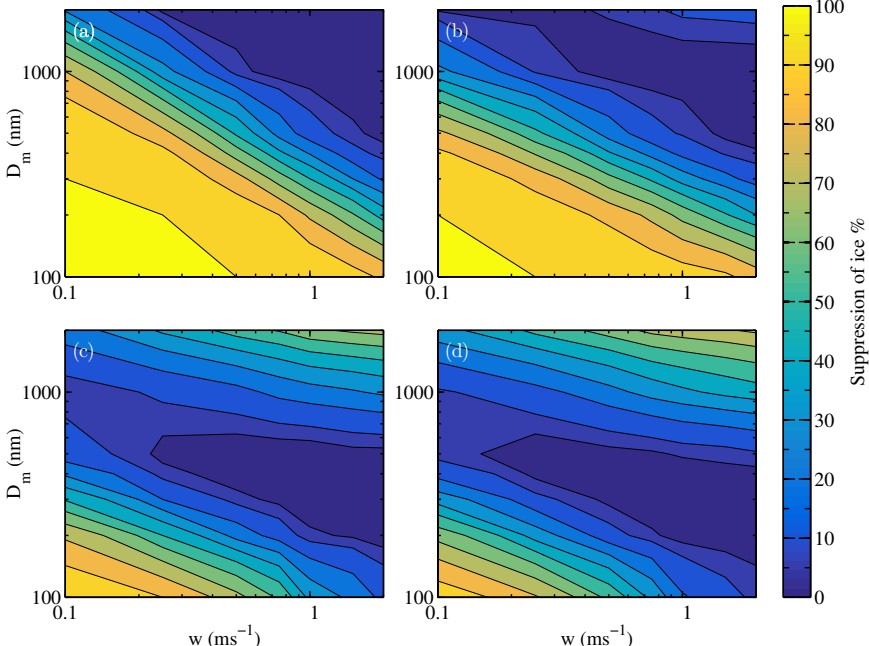

**Figure 5.** Results from 196 pairs of *low CCN* and *high CCN* model simulations with initial conditions as detailed in Tables 3 and 4. For each of the four soluble fractions of INP the updraft velocity ranged between 0.1 and 2 ms$^{-1}$ and the diameter of the INP varied between 100 and 2000 nm. The ice crystal number concentration in every simulation was taken at -30$^o$C. Contours show the percentage less ice that formed in the *high CCN* case compared to the *low CCN* case. The criteria for heterogeneous freezing in these results is only activated drops can freeze.

with INPs that include a soluble fraction, are reduced compared to equivalent simulations at lower updrafts. This highlights that kinetic limitations to growth and the competition between the INPs for water vapour exists in the absence of CCN. The introduction of CCN in the *high CCN* case further reduces the ice crystal number concentration by reducing the supersaturation in the parcel, thus the number of INPs that can activate and freeze.

5     The suppression at high updraft velocities and large INP median diameters is enhanced when the ice crystal concentrations between the *low CCN* and *high CCN* cases are compared at higher temperatures. Supplementary Figures S7 and S8 are similar to Fig. 5. however the ice crystal number concentrations are compared at -15$^o$C and -20$^o$C respectively. The greater suppression seen with increasing temperature is due the kinetic limitations to growth of large particles. The simulation time required to reach -20$^o$C and -15$^o$ is less than that required to reach -30$^o$C therefore INPs have less time to grow, activate and freeze. Slightly less

10   suppression is seen at lower updrafts and smaller diameters of INPs, bottom left corners of panels in Supplementary Figs. S7. and S8. This is because at warmer temperatures the number concentration of ice in the *low CCN* case is still increasing with

©c Author(s) 2017. CC BY 4.0 License.




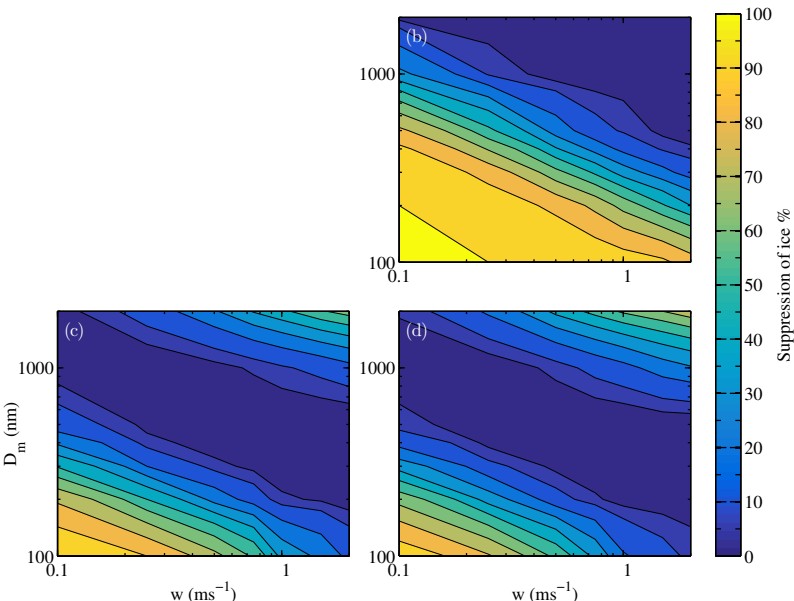

**Figure 6.** Same as Figure 5 however criteria for heterogeneous freezing in these results is $A_w$.

decreasing temperature however in the *high CCN* case most of the INPs that have activated and therefore able to freeze have already frozen.

Figure 6 shows results for the same simulations as in Fig. 5. (and Fig. 7.) however the criteria for freezing is $A_w$. Panel a is missing for all figures where the criteria for freezing is $A_w$, (Fig. 6., Supplementary Figs. S9., S10. and S14). This is because

the water activity of all INPs would be equal to 1 in this case, as there is no soluble fraction, meaning $\kappa = 0$, see Eq. (3). This results in no suppression as INPs always have a water activity greater than the threshold 0.9999. Results for the suppression of ice using the criteria for freezing of $A_w$ shown in panels b, c and d, Fig. 6., are similar to those of *Activated Only*, Fig. 5., in Regime 1. However Fig. 6. shows less suppression in Regime 2, panels c and d, compared to *Activated Only*, Fig. 5.

The simulations show in Fig. 5. and Fig. 6. were repeated using the $M_{cw}$ criteria for freezing. The results of these simulations

are shown in Fig. 7. Panels a and b in Fig. 7. are very similar to the corresponding panels in Fig. 5. and Fig. 6. Panels c and d, soluble fractions of INP 25 % and 50 % respectively, Fig. 7., show some suppression in Regime 1 and no suppression in Regime 2.

The magnitude of the suppression of ice formation for insoluble and slightly soluble INPs, panels a and b Fig. 5., Fig. 6. and Fig. 7., is very similar. This is because the threshold mass of condensed water required for the freezing of particles with very

low to no soluble fractions, defined by the three criteria, *Activated Only*, $A_w$ and $M_{cw}$, are very similar. At higher INP soluble fractions, panels c and d, the mass of water required for a particle to activate or reach the threshold $A_w$ increases. However the





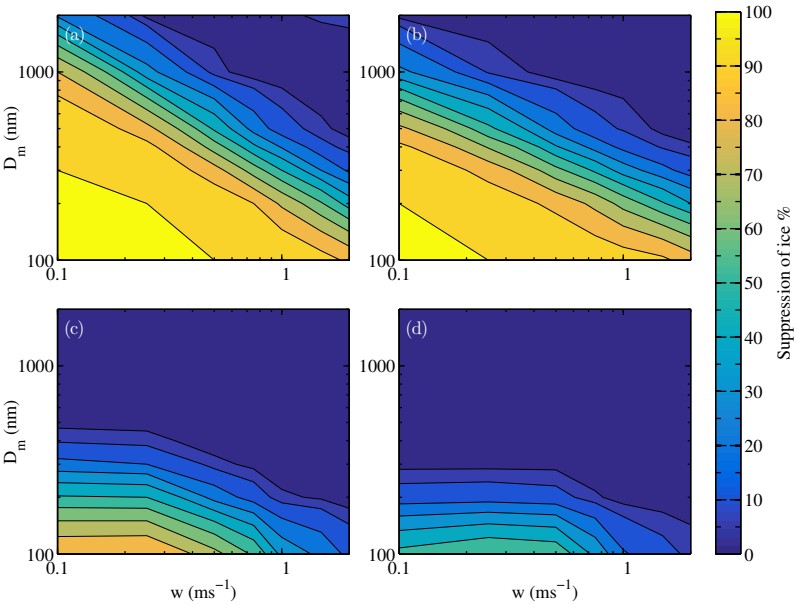

**Figure 7.** Same as Figures 5 and 6 however the criteria for heterogeneous freezing in these results is $M_{cw}$.

mass required to reach $M_{cw}$ remains the same as it is for low soluble fractions as the calculation of $M_{cw}$ does not depend on particle composition. This results in little to no suppression in panels c and d Fig. 7. for the $M_{cw}$ criteria. For particles with higher soluble fractions the mass of water required for activation and to reach $A_w$ is similar for particle sizes below about 800 nm, see Supplementary Fig. S16. This explains the similarities in the magnitude of suppression seen in Regime 1 in panels c

and d of Fig. 5. and Fig. 6. Less suppression in Regime 2 is seen for the $A_w$ criteria compared to the *Activated Only*. This is because at particle sizes >~800 nm the mass of water required for activation is larger than that that is required to reach $A_w$ for particles with significant soluble fractions.

Simulations, the results of which are shown in Fig. 5., Fig. 6. and Fig. 7., were repeated with a representation of desert dust instead of K-Feldspar. The results from these simulations are presented in Supplementary Figs. S13., S14. and S15. and are

10 very similar to the results found for K-feldspar except with slightly more suppression seen in Regime 2 for INPs with some soluble fraction (panels b, c and d). The ice nucleating ability of desert dust follows Niemand et al. (2012). The representation of desert dust used here is less ice active compared to K-feldspar.

The effect of soluble material in solution on freezing is implicitly taken into account by the $A_w$ criteria for freezing through $\kappa$ in Eq. (3). Higher values of $\kappa$, which represent higher soluble fraction of INP, result in a lower $a_w$ compared to a particle

with the same dry and wet size but lower soluble fraction. Therefore a particle with a high $\kappa$ value will require more condensed water in order to reach the threshold than a particle with a low $\kappa$ value.





The $M_{cw}$ criteria for freezing does not take into account the effect of soluble material on freezing. There is no dependence in the equation for $M_{cw}$, Eq. (7), on particle type, only particle size. However, as is the case with $A_w$ and all other freezing criteria, once the criteria has been reached, the ability of a particle to freeze is determined by $n_s$ which is dependent on the mass fraction of ice active compounds. The higher the soluble mass fraction of an INP the lower $n_s$ will be therefore the less

likely an INP will be to freeze.

In order to explicitly assess the effect of soluble species on the freezing of a drop, a parameterisation following Diehl and Wurzler (2004) was included in the model which includes the effect of solution on freezing. This parameterisation uses Koop et al. (2000) parameterisation for homogeneous freezing which describes that the freezing of a solution drop is only dependent on $a_w$, and that homogeneous freezing begins when the freezing rate, $J$, is larger than 1 cm$^{-3}$s$^{-1}$, (Pruppacher and Klett,

1997). Supplementary Fig. S5. shows the fit used here for the freezing temperature of a solution, $T_{fs}$ when $log J = 0$ cm$^{-3}$ s$^{-1}$. The effect of soluble compounds, present in a drop, on freezing is taken into account by calculating the freezing point depression,

$$T_{dep} = T + \Delta T \tag{9}$$

Where $\Delta T$ is the difference between $T_{fs}$ and the freezing temperature when $a_w = 1$, i.e. when no solute is present. $T_{dep}$ is

then used to calculate the number of ice active sites present in a drop using $n_s(T_{dep})$. For $a_w < 1$ $T_{dep}$ will be higher than the actual temperature at the given model time step therefore $n_s$ will be lower when solutes are present. Less suppression of ice formation was found when the effect of a freezing point depression caused by the presence of soluble compounds was taken into account in the model compared to results for $M_{cw}$, Supplementary Fig. S6. The maximum suppression seen in Supplementary Fig. S6. is around 20% which occurs in the lowest soluble fraction case, 1% soluble fraction. Our results show that when there

is competition for water vapour from CCN particles, INPs with a soluble component are able to compete effectively for water vapour due to their increased kappa value and grow sufficiently dilute that there is no freezing point depression.

## 6 Conclusions

The competition for water vapour between INPs and CCN can result in the suppression of ice formation if it is assumed that only activated drops can freeze. Such an assumption has been made in the literature for *immersion mode* freezing, (Hoose et al.,

2010b; Pruppacher and Klett, 1997). The suppression effect is greatest in low updraft conditions where the INPs have small diameters and are mostly insoluble. Two 'regimes' of the suppression effect can be seen in the results of our sensitivity study. The first occurs at low updrafts and small INP diameters. Here INPs are in competition with CCN for available water vapour which results in them not receiving enough water to be able to activate and freeze. The second regime occurs at higher updrafts and INPs with a significant soluble fraction and large median diameter. Here the INPs act as giant CCN resulting in them being

in competition with themselves as well as the CCN for water vapour.

Significantly less suppression is seen, and is confined to INPs with low soluble fractions and diameters less than 400 nm, if the criteria for heterogeneous freezing is that an INP must receive a threshold mass of condensed water and that mass is

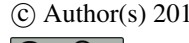


small compared to its activated size. Such criterion were used here to accurately simulate ice formation during cloud chamber experiments. Other criteria for heterogeneous freezing defined in the model, such as a drop must be activated or water saturation must be reach, failed to consistently reproduce the ice crystal number concentrations measured in chamber experiments. This indicates that further investigation is required into the criteria for heterogeneous freezing as here we have shown it can be the

5   difference ice formation and no ice formation. Further work will include an investigation into how this suppression effect may manifest its self in large scale weather and climate models.

*Competing interests.*   The authors declare that there are no competing interests.

*Acknowledgements.*   This work was funded by a NERC studentship, grant number NE/L501591/1. Funding for cloud chamber work was provided under CCN-Vol, grant number NE/L007827/1. E. Simpson would like to thank James Dorsey and Angela Buchholz for their help

10  and support during the cloud chamber experiments.





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
