# Peer review of "Competition for water vapour results in suppression of ice formation in mixed phase clouds"

_Atmospheric Chemistry and Physics, 2017_

## Referee Comment (RC1) · Anonymous Referee #2 · 12 Sep 2017

This manuscript presents laboratory experiments and parcel model simulations of ice nucleation at temperatures mixed phase clouds are observed. The experiments simulate expansion cooling in a chamber, in which ice formation on K-feldspar particles (INP) was measured at low and high ammonium sulfate (CCN) concentrations, respectively. A parcel model is then used to simulate the chamber experiments.

The parcel model simulates droplet activation for insoluble as well as soluble aerosols, and simulates ice nucleation based on ice active sites per unit surface area. The authors used four different criteria for initiation of heterogeneous freezing. Two of the criteria (threshold mass of condensed water, threshold water activity) that were derived by fitting the chamber results are non-traditional.

The parcel model is then used to simulate ice formation for a range of vertical veloci-

ties and INP size distributions. Model results are compared between low and high CCN simulations, which suggests that competition for water vapour may lead to suppression of ice formation in mixed phase clouds. This conclusion is not supported by the chamber experiments (Line 14, Page 9). However, the experiments and model simulations are still interesting and are appropriate for publication in ACP. For these reasons a major revision is recommended.

Major Comments:

1. There is a lack of discussion on possible reasons why a suppression of ice formation by high CCN was not observed in the chamber experiments or what assumptions made in the parcel model may be inaccurate that led to the suppression.

2. If "a minimum depth" of aqueous shell is required for ice nucleation (page 6, line 28), then the critical water mass required for freezing should be proportional to the K-feldspar particle surface area, not the volume as shown in Eq. (7).

3. A threshold water activity may not be a good criterion, as indicated by the narrow range (0.9997 to 0.99994) of fitting results. Furthermore, ice nucleation has been observed to occur at lower water activities, although decreasing rapidly as the activity decreases (e.g., Knopf and Alpert, 2013).

4. The two new criteria (threshold water mass and threshold water activity) were derived from experiments with INPs of a specific size distribution. Would the same values be estimated when the INP size distribution is changed or the soluble mass fraction is changed in the chamber experiments? The estimated criteria are used in Section 5 for various size distributions and soluble mass fractions.

Minor Comments:

1. Parcel model simulations were made for a range of INP size distributions and vertical velocities, using the four different freezing criteria (shown in Figs. 5, 6, 7, S6-S15). The results demonstrate the suppression of ice nucleation when CCN is numerous and

droplet growth effectively competes for water vapor. The interpretation is reasonable for Regime 1 (small size and low vertical velocity). But Lines 5-8 on Page 15 may need some clarification:

"In the second regime, Regime 2, towards the top right hand corners of panels c and d in Fig. 5, INPs have a significant soluble fraction that allows them to act as giant CCN at large diameters, towards 2 um. This results in a suppresion of ice formation because the INPs are in competition with themselves as well as the CCN for available water vapour."

Is it possible that the suppression results from earlier ice nucleation (i.e., at higher T) with larger INPs and subsequent vapour deposition to ice enhances the effect of CCN?

Similar clarification is suggested for Lines 7-11 on Page 16:

Is "kinetic limitations to growth of large particles" meaning more rapid growth of ice particles?

"This is because at warmer temperatures..." the WBF process is also more effective at lower updrafts?

2. Line 28 on Page 2. "do not take competition for water vapor into account as it has already been corrected for" seems to be saying the opposite.

3. Move the text on Lines 1-15 on Page 14 (description of FHH, the hygroscopicity parameter, and soluble fraction) to between Lines 27 and 28 on Page 4, as a new paragraph. Start a new paragraph at "Freezing rates of INPs ..." (Line 28).

4. What is the soluble mass fraction and hygroscopicity parameter for K-feldspar particles used in the chamber experiments and in the parcel model? Give the values near Line 23 on Page 4 or in Table 2 on Page 12.

5. Figure 2(b) on Page 10. The peak concentrations slightly beyond 250 seconds appear to be spurious.

6. Table 2 on Page 12. One modal size distribution is given for K-feldspar particles in Table 2. But a two-modal size distribution is shown in Figure S4. Need to explain why the fine mode is neglected.

7. Lines 10-11 on Page 13. "INPs with four different soluble fractions" could also be given earlier in model description (see minor comment 3).

8. Lines 29-30 on Page 19. "the INPs act as giant CCN" is not as effective as ice formation and subsequent vapor-to-ice deposition for INPs in competition with themselves.

9. Lines 4-5 on Page 20. This statement is confusing.

10. Figure S1. Why are the curves for frozen fraction not monotonic as a function of INP diameter (in units of nm)?

11. Figure S16. Is the unit for Dp micron rather than nm? Also the caption is not clear: what are the chamber conditions (e.g., soluble mass fraction?), what is the difference between (a) and (b)?

12. Typos. "approached" on Line 9, Page 4. "location long" on Line 5, Page 8. "where" on Line 8, Page 13. "reach" on Line 3, Page 20. "its" on Line 6, Page 20.

References:

Knopf, D.A., and Alpert, P.A.: A water activity based model of heterogeneous ice nucleation kinetics for freezing of water and aqueous solution droplet, Farad. Discuss., 165, 513-534, 2013.
* * *

---

## Referee Comment (RC2) · Anonymous Referee #1 · 15 Sep 2017

The core idea in this paper is that high concentrations of CCN can inhibit the formation of ice through a Wegener-Bergeron-Findeisen (WBF) type effect, where soluble particles activate and reduce the local supersaturation enough that it prevents the ice nucleating particle from acquiring enough water to support formation of a critical embryo. This is an intriguing idea, but there are some serious gaps in its presentation in this paper. I am not opposed to publication, but some of the points below need to be addressed.

The experiments do not contribute much to the paper. The sentence on page 9 (line 13) is the clearest statement of this. "However no suppression of ice was observed." While I admire the authors' honesty in explicitly stating this, it calls into question the premise of having the experiments in the paper at all. The next statement in the manuscript

is an explanation of why they didn't see the suppression and how it is consistent with the results from simulations. If the model doesn't show suppression of ice formation in this parameter range, why not run experiments in the range where the model **does** show suppression? That might not be possible, but if it isn't that should be stated and explained.

Once it is established that the experiments do not show the paper's core idea, the purpose seems to be a refinement of nucleation parameterizations of the various dusts that were used in the study. I do not find these results convincing enough to justify using any of the criteria listed in Table 1. (There are criteria in Table 1 that I find compelling. See next paragraph.) If a better curve fit is the goal, why not just modify the original parameterization?

In the end, the claim that freezing only proceeds with some critical amount of water is reasonable. There is evidence to support it. For example, Sanz et al have calculated the size of the critical embryo as a function of supercooling. Li's work also indicates that the nucleation rate in constrained volumes can differ from the bulk. But results from a cloud chamber are unlikely to experimentally confirm these findings. There are too many other explanations that are also likely that could be invoked. (Please see the question concerning heat leakage into the chamber in **Miscellaneous**.)

Suppression of ice formation is more clearly supported by the simulations. I agree that a minimum amount of water is needed (setting aside deposition nucleation, for the moment), even if I am not convinced that the measurements show this. There is an aspect of the simulations that is not discussed as well as it should be though. There is an implicit assumption that the ice nucleating particles are submicron. While most of the ice nucleating particles that have been measured in the atmosphere are submicron, this is, in part, an artifact of the measurement technique. For example, in most Continuous Flow Diffusion Chamber measurements, the larger particles are intentionally excluded because the discrimination between liquid water and ice is done on the basis of size. There are results showing that particles larger than a micron are an

appreciable fraction of atmospheric ice nucleating particles (Mason et al 2016). If you consider ice nucleating particles in that size range, the argument based on competition for vapor starts to break down. The critical supersaturation for a 1 micron particle with a kappa value of 0.01 is 0.04%, which is comparable to an ammonium sulfate particle of 250 nm diameter. With those values in mind, I do not find CCN "outcompeting" ice nucleating particles as compelling.

In summary, the idea that ice formation could be suppressed because of competition between numerous CCN and ice nucleating particles is an intriguing one, but the chain of reasoning presented here should be tightened.

**Miscellaneous**

Do a global search for "preform". This occurs in several places in the paper, and I am almost certain that it should be "perform".

Similarly, search for "modelling". Delete one of the "l"s from that word.

Page 8, line 3: "were" not "where"

The reference to Rogers and Yau on page 7 (line 18) is not needed. If you are going to cite it, please at least provide a chapter or section. That book covers a lot of topics.

The reference to Kumar 2009 on page 14, line 12 should be in parentheses.

Is there a leakage of heat into the experimental chamber? There's a substantial difference between thermocouple 1 and thermocouple 8 by the time you get to 250 seconds after expansion starts. Could that be part of the reason you see the decrease in ice crystal concentrations as a function of time? (I would have expected to see ice crystal concentrations increasing or at least staying constant since the counters are at the bottom of the chamber and crystals from above are continuously falling into the field of view.)

**References**

Sanz, E., Vega, C., Espinosa, J.R., Caballero-Bernal, R., Abascal, J.L.F., Valeriani, C., 2013. Homogeneous Ice Nucleation at Moderate Supercooling from Molecular Simulation. J. Am. Chem. Soc. 135, 15008–15017. doi:10.1021/ja4028814

Li, T., Donadio, D., Galli, G., 2013. Ice nucleation at the nanoscale probes no man's land of water. Nature Communications 4, 1887. doi:10.1038/ncomms2918

Mason, R.H., Si, M., Chou, C., Irish, V.E., Dickie, R., Elizondo, P., Wong, R., Brintnell, M., Elsasser, M., Lassar, W.M., Pierce, K.M., Leaitch, W.R., MacDonald, A.M., Platt, A., Toom-Sauntry, D., Sarda-Estève, R., Schiller, C.L., Suski, K.J., Hill, T.C.J., Abbatt, J.P.D., Huffman, J.A., DeMott, P.J., Bertram, A.K., 2016. Size-resolved measurements of ice-nucleating particles at six locations in North America and one in Europe. Atmos. Chem. Phys. 16, 1637–1651. doi:10.5194/acp-16-1637-2016

---

## Author Comment (AC1) · 5 Mar 2018

Author's Note:

After submission of the manuscript it came to light that the temperature measurements from the chamber experiments presented, exhibited a lag, with a time constant of around 60 seconds, which has now been corrected, see the appendix in Frey et al (in review 2018) for a detailed discussion.

The revised manuscript is substantially changed as a result of the revised temperature measurements as well as the reviewer comments. The major changes that follow from the temperature measurements are that the new criteria for freezing of a threshold water activity can no longer be meaningfully constrained with our limited number

of experiments. The revised temperature measurements have also increased the uncertainty in estimation of the initial RH values used in model simulations of chamber experiments. As a result a range of RH values have been used when modeling chamber experiments, which represent the uncertainty in the initial RH. Our original paper stated that the new criterion for freezing, threshold mass of water, agreed better with chamber observations than the criterion that only activated drops can freeze. In our revised manuscript we contend that both criteria can represent ice formation in our chamber experiments to the same degree of accuracy. More detailed lognormal fits to the aerosol present in the chamber were also calculated in order to further improve model simulations of chamber experiments, and are included in Supplementary Table 1. These new fits do not significantly impact the results of the paper however have been included for completeness.

Changes to the manuscript following the correction of temperature measurements are confined to the 'Chamber Experiments' section of the paper. Results presented in other sections of the manuscript remain unchanged.

Over all the main themes of the paper have not changed.

The changes made in response to the reviewer comments are included in our point-by-point response below.

Author's response to Anonymous Referee #1 comments:

[Format = 'quoted comment', author's response, changes made]

Comment 1: 'The experiments do not contribute much to the paper. The sentence on page 9 (line 13) is the clearest statement of this. "However no suppression of ice was observed." While I admire the author's honesty in explicitly stating this, it calls into question the premise of having the experiments in the paper at all. The next statement in the manuscript is an explanation of why they didn't see the suppression and how it is consistent with the results from simulations. If the model doesn't show suppression of

ice formation in this parameter range, why not run experiments in the range where the model does show suppression? That might not be possible, but if it isn't that should be stated and explained.'

The reviewer draws attention to the fact that a clear description of the purpose of the chamber experiments has not been given as well as an explanation as to why we were unable to conduct chamber experiments in the parameter space where suppression is predicted by the model. We will address both these points in detail here.:

We strongly disagree with the reviewer and do not subscribe to the contention that a positive identification of the phenomenon is the only worthwhile result. The null result importantly confirms one part of the hypothesis – that the competition for water vapour should not be observed under the conditions accessible to our chamber. The area of the parameter space where the suppression effect is predicted to be greatest is either with small sized INPs under low updrafts or with large INPs, with a significant soluble fraction, under high updraft conditions. With our current set-up we are only able to generate high pressure drop rates (analogous to high updraft conditions) in the chamber. We were also not able to produce INP particles with a known soluble mass fraction. With these two experimental limitations it was not possible to conduct experiments in which suppression is likely to be seen. Therefore experiments were conducted in the parameter space least likely to see suppression; high updrafts, moderate to large INP sizes (0.35 – 1.5 micro-meters) which are composed only of insoluble dust. Chamber results showing no suppression agree with model predictions.

Changes to the manuscript have been made Page 3, Lines 13 - 20 - 'The two main objectives of this work are to provide experimental evidence that our hypothesis is valid and to investigate the sensitivities of the process. Section 2 provides a description of the model used, as well as a detailed description of the freezing criteria we employ in the model. Section 3 details the methodology and results of a series of cloud chamber experiments designed to confirm our hypothesis. Due to constrains of our chamber set-up our experiments probe an area of the parameter space where suppression is

not predicted to occur. The results from chamber experiments are also used in a comparison of different freezing criteria and provide a 'proof-of-concept' for our two new criteria for freezing. Section 4 provides a demonstration of the suppression effect in model simulations and Sect. 5 explores the sensitivities to the suppression effect. Finally a summary of the overall findings of this work is given in Sect. 6.'

Page 10 Lines 10 – 14 - 'Ammonium sulphate aerosol was included in some of the expansions (Supplementary Fig. 3. panels c through f) in order to provide a source of CCN. As hypothesized, no suppression of ice was observed. In the area of the parameter space were our chamber experiments are conducted, i.e. insoluble particles with median mode diameters of 0.35 micrometer and 1.5 micrometer and moderate to high updraft velocities, similar to the pressure drop rates in the chamber, little suppression is found in model simulations when using either of the three criteria for freezing compared in Sect. 5.'

Comment 2: 'Once it is established that the experiments do not show the paper's core idea, the purpose seems to be a refinement of nucleation parameterizations of the various dusts that were used in the study. I do not find these results convincing enough to justify using any of the criteria listed in Table 1. (There are criteria in Table 1 that I find compelling. See next paragraph.) If a better curve fit is the goal, why not just modify the original parameterisation?'

A better curve fit is not the goal of the paper. We recognise that the experiments presented in this study are insufficient for the development of new parameterisations for ice nucleation. However that is not the purpose of this work. The purpose of this work is to investigate a new cloud process and explore the potential sensitivities to that process. The ice nucleation parameterisation used in our model follows the work of Connolly et al (2009) and Niemand et al (2012) and is well established. The criteria listed in Table 1 are not parameterisations for ice nucleation, instead they are a means to define when sufficient water is present on a INP in order to allow freezing to occur, see Page 6 lines 4 – 22. We suggest two new, physically based, criteria for heterogeneous freezing

and compare them to our limited experimental results in order to demonstrate their potential usefulness. These new criteria, and one established criteria from the literature, are used in the sensitivity study of the suppression effect in order to demonstrate the significant impact heterogeneous freezing criteria can have on ice number concentration. The use of the new criteria for freezing in our model simulations is to highlight a significant sensitivity to the suppression of ice formation, that is currently somewhat unconstrained.

Comment 3: 'In the end, the claim that freezing only proceeds with some critical amount of water is reasonable. There is evidence to support it. For example, Sanz et al have calculated the size of the critical embryo as a function of supercooling. Li's work also indicates that the nucleation rate is constrained volumes can differ from the bulk. But results from cloud chamber are unlikely to experimentally confirm these findings. There are too many other explanations that are also likely that could be invoked. (Please see the question concerning heat leakage into the chamber in Miscellaneous.)'

We agree with the reviewers insightful appraisal and for pointing us at the literature in this area. We also think that the contention that freezing only proceeds with some critical amount of water is reasonable, leading to our testing of the model response to this criteria. It is not our intention to discriminate between bulk ice nucleation and freezing in constrained volumes such as drops.

Comment 4: 'Suppression of ice formation is more clearly supported by the simulations. I agree that a minimum amount of water is needed (setting aside deposition nucleation, for the moment), even if I am not convinced that the measurements show this. There is an aspect of the simulations that is not discussed as well as is should be though. There is an implicit assumption that ice nucleating particles are submicron. While most of the ice nucleating particles that have been measured in the atmosphere are submicron, this is, in part, an artifact of the measurement technique. For example, in most Continuous Flow Diffusion Chamber measurements, the larger particles are intentionally excluded because the discrimination between liquid and ice is done on

the basis of size. There are results showing that particles larger than a micron are an appreciable fraction of the atmospheric ice nucleation particles (Mason et al 2016). If you consider ice nucleating particles in that size range, the argument based on competition for vapor, starts to break down. The critical size of a 1 micron particle with a kappa value of 0.01 is 0.04%, which is comparable to an ammonium sulphate particle of 250 nm diameter. With those values in mind, I do not find CCN "outcompeting" ice nucleating particles as compelling.'

There is no implicit assumption in our modeling work that INPs are submicron. In the chamber experiments there are two modes of INP present, with median diameters 0.35 and 1.5 microns, and in the sensitivity simulations (Figures 5, 6 and 7) the INP model diameter varies from 0.1 to 2 microns. Therefore supermicron INPs are included, as are submicron INPs.

In our simulations the population of CCN is significantly larger than the population of INPs, therefore a greater mass of water is taken up by the CCN population than INPs. Also the growth of a 1 micron particle towards its critical diameter will be much slower than the growth of a 250 nm particle due to kinetic limitations to growth which affect particles with large diameters (Chuang et al, 1997). So while the critical supersaturations of these two particles maybe similar, the smaller particle will activate, and subsequently experience rapid growth, before the larger particle is able to activate, thus the CCN create a sink for water vapour which will reduce the ambient supersaturation and prevent further activation of large particles.

Changes to the manuscript Page 12 Lines 9 - 10 - 'The number of CCN is significantly greater than the number of INPs in the 'high CCN' case therefore a greater mass of water is taken up by the CCN population than the INPs.'

Miscellaneous

'Do a global search for "preform". This occurs in several places in the paper, and I am almost certain that it should be "perform".'

All instances of the word "preform" have been corrected to "perform".

'Similarly, search for "modelling". Delete one of the "l"s from that word.'

All instances of the word "modelling" have been changed to "modeling".

'Page 8, line 3: "were" not "where".'

This has been changed.

'The reference to Rogers and Yau on page 7 (line 18) is not needed.'

This reference has been removed.

'The reference to Kumar et al 2009 on page 14, line 12 should be in parentheses.'

This has been changed.

'Is there a leakage of heat into the experimental chamber? There's a substantial difference between thermocouple 1 and thermocouple 8 by the time you get to 250 seconds after expansion starts. Could that be part of the reason you see the decrease in ice crystal concentrations as a function of time? (I would have expected to see ice crystal concentrations increasing or at least staying constant since the counters are at the bottom of the chamber and crystals from above are continuously falling into the field of view.)'

Yes there is a transfer of heat from the walls to the gas in the chamber. There is a decrease in ice concentration because the measurement is in number per cm3 of air and there is a change in air density during the experiment. This decrease in ice crystal number can also be seen in the model simulations of chamber experiments in Figure 3, where the unit conversion between number per kg of air to number per cm3 has been made resulting in a slight decreasing trend in ice crystal concentrations.

Author's response to Anonymous Referee #2 comments:

Major Comments:

Comment 1: 'There is a lack of discussion on possible reasons why a suppression of ice formation by high CCN was not observed in the chamber experiments or what assumptions made in the parcel model may be inaccurate that led to the suppression'

Anonymous Referee #1 also raised the point that there is a lack of discussion on the purpose of the chamber experiments, therefore we refer Referee #2 to our response to Comment 1 of Referee #1.

Regarding inaccurate assumptions made in the model that could lead to a suppression effect, perhaps the fact that it is assumed that koehler theory still applies at temperatures below 0oC could lead to inaccurate results. However this is a widely made assumption and, to the Author's knowledge, its validity has not been investigated.

Comment 2: 'If "a minimum depth" of aqueous shell is required for ice nucleation (page 6, line 28), then the critical water mass required for freezing should be proportional to the K-feldspar particle surface area, not the volume as shown in Eq. (7).'

The surface of a K-feldspar particle is likely not to be spherical, with many facets leading to water condensing into irregular pools of water on the particle's surface, in which ice can nucleate. The depth of these pools is therefore not in direct relation to the surface area of the drop nor the drop's volume, however will likely be somewhere in between.

Changes to the manuscript – Page 7 Lines 16 - 21 - ' The surface of an ice nucleus is not typically spherical, (Rogers et al 2001) and will instead have many facets, leading to water condensing into irregular pools of water on the particle's surface, in which ice can nucleate. This means that the depth of the liquid layer on the INP's surface will not be in direct relation to the particle's surface area nor the drop's volume, however will be somewhere in between. We have chosen to calculate the threshold water mass in relation to particle volume, as it is not possible to know the exact morphology of the particles.'

Comment 3: 'A threshold water activity may not be a good criterion, as indicated by the narrow range (0.9997 to 0.99994) of fitting results. Furthermore, ice nucleation has been observed to occur at lower water activities, although decreasing rapidly as the activity decreases (e.g. Knopf and Alpert, 2013).'

We disagree with the reviewer and believe that the threshold water activity is as good a criteria for freezing as a critical volume/mass of water as it is just a re-expression of the latter. We adjust the threshold water activity between 0.9997 and 0.99994 which represents a 70% difference in particle wet diameter and corresponding factor of 5 difference in particle volume.

In light of corrected temperature measurements in the chamber it is not possible, with the current observations, to meaningfully constrain the value of a threshold water activity for freezing. This is due to the significant uncertainty in initial RH values.

Comment 4: 'The new criteria (threshold water mass and threshold water activity) were derived from experiments with INPs of a specific size distribution. Would the same values be estimated when the INP size distribution is changed or the soluble mass fraction is changed in the chamber experiments? The estimated criteria are used in Section 5 for various size distributions and soluble mass fractions.'

If the theory that a threshold mass of water or threshold water activity required for freezing to proceed exists then the same threshold should exist over all size distributions. However it is likely that with further experiments, using different particle size distributions and soluble mass fractions, different values for the threshold mass and water activity will be calculated, as our current set of experiments is limited. The use of the values we currently have over various size distributions is to demonstrate that the suppression effect is very sensitive to the freezing criteria chosen in the model. And to highlight the need to better constrain the requirements for heterogeneous freezing.

Minor Comments:

Minor Comment 1: 'Parcel model simulations were made for a range of INP size distributions and vertical velocities, using the four different freezing criteria (shown in Figs. 5, 6, 7, S6-S15). The results demonstrate the suppression of ice nucleation when the CCN is numerous and droplet growth effectively competes for water vapour. The interpretation is reasonable for Regime 1 (small size and low vertical velocity). But lines 5-8 on Page 15 may need some clarification:

"In the second regime, Regime 2, towards the top right hand corners of panels c and d in Fig. 5, INPs have a significant soluble fraction that allows them to act as giant CCN at large diameters, towards 2 microns. This results in a suppression of ice formation because the INPs are in competition with themselves as well as the CCN for available water vapour."

Is it possible that the suppression results from earlier ice nucleation (i.e., at higher T) with larger INPs and subsequent vapour deposition to ice enhances the effect of CCN?

Similar clarification is suggested for Lines 7-11 on Page 16:

Is "kinetic limitations to growth of larger particles" meaning more rapid growth of ice particles?

"This is because at warmer temperatures ..." the WBF process is also more effective at lower updrafts?'

The timing of ice formation is similar in both high and low CCN cases, the increase in suppression in Regime 2 is due to the increase in competition for water vapour from the large INPs.

'Kinetic limitations to growth of larger particles' refers to the fact that the growth rate of larger particles is less than that of smaller particles (see Chuang et al 1997). The ability of an INP to obtain sufficient water mass, in order for freezing to occur, is limited by time. At shorter time scales less INPs are therefore able to freeze.

The reason why less suppression is seen in Regime 1 at warmer temperatures is be-

cause the number concentration of ice in the low CCN cases is less than at lower temperatures. However in the high CCN cases the number concentration of ice at the warmer and colder temperatures are similar. This is because the INP only have a limited amount of time to grow into liquid drops before the competition for water vapour by the CCN limits the supersaturation, and thus slows the growth of the INPs.

Changes to the manuscript, Page 15 Line 12, Page 16 Lines 1 - 8 - 'In the second regime, Regime 2, towards the top right hand corners of panels c and d in Fig. 5., INPs have a significant soluble fraction that allows them to act as giant CCN at large diameters, towards 2 microns. This results in a reduction of ice formation in high CCN simulations and, to a lesser extent, in low CCN simulations. The reason for this reduction in ice formation is that there is increased competition for water vapour due to the INPs acting as giant CCN. Another contribution to the suppression effect in Regime 2 is the higher updraft velocities. The time taken to reach -30oC in simulations with high updrafts is less than in simulations with low updrafts. The growth rate of large aerosol particles is less than that of small aerosol particles due to kinetic limitations to growth, (Chuang et al 1997). This means that with less time, fewer INPs are able to grow sufficiently in order for freezing to occur.'

Page 16, Lines 17 - 25 - 'At warmer temperatures the number concentration of ice in the low CCN cases is less than at colder temperatures. However in high CCN cases the number concentration of ice at -15C, -20oC and -30oC is similar. INPs in the ow CCN cases have a greater potential to grow sufficiently in order to freeze, as a relatively high supersaturation is maintained for a larger proportion of the simulation time due to limited competition for water vapour. In high CCN simulations the supersaturation is rapidly reduced, meaning that the growth period of INPs is limited. The period of time in which INPs are able to significantly grow occurs before approximately -15C in high CCN cases therefore the ice number concentration is similar at -15oC, -20oC and -30oC. However in the low CCN cases a relatively high supersaturation is maintained beyond -15oC allowing INPs to continue growing at lower temperatures and the ice

crystal number concentration to continue to increase. Thus ice crystal concentrations in the low CCN cases is significantly higher at lower temperatures. '

Minor Comment 2: 'Line 28 on Page 2. "do not take competition for water vapour into account as it has already been corrected for" seems to be saying the opposite.'

Clarification: The formulation of the parameterisation assumes that particles are not prevented from freezing due competition for water vapour. Competition for water vapour in experiments is corrected for, so that the parameterisation can be calculated assuming all particles can grow into drops.

Changes to the manuscript, Page 2 Lines 26 - 28 - 'This means that their formulation of INP parameterisations do not take competition for water vapour into account as they have been calculated assuming all particles can grow into drops.'

Minor Comment 3: 'Move the text on Lines 1-15 on Page 14 (description of FHH, the hygroscopicity parameter, and soluble fraction) to between Lines 27 and 28 on Page 4, as a new paragraph. Start a new paragraph at "Freezing rates of INPs ..."(line 28).'

The suggested changes have been made, Page 5, Lines 8 – 20.

Minor Comment 4: 'What is the soluble mass fraction and hygroscopicity parameter for K-feldspar particles used in chamber experiments and in the parcel model? Give the values near Line 23 on Page 4 or in Table 2 on Page 12.'

Changes to the manuscript, Page 5, Lines 4 - 6 - "In the simulations of chamber experiments K-feldspar particles are given a 1% soluble mass fraction made up of ammonium sulphate. This results in K-feldspar particles having a kappa value of 0.0061 in the model."

Minor Comment 5: 'Figure 2(b) on Page 10. The peak concentrations slightly beyond 250 seconds appear to be spurious.'

The large increase in particle number concentration seen in Figure 2(b) towards the

end of the experiment is due to pieces of ice breaking off the pump values as they are closed at the end of the experiment. Frost forms on the chamber walls and the pump valves. When the pumps are switched on and off pieces of frost break off. This can be seen in the measurements as high concentrations of particles at the beginning and end of experiments.

Changes to the manuscript, Page 11 Line 7 - 10 - 'The peak in concentration in Fig. 1a slightly beyond 250 seconds is due to ice breaking off the valves connected to the pumps when the pumps are switched off. Frost forms on the chamber walls and the pump valves. When the pumps are switched on and off pieces of frost break off. This can be seen in the measurements as high concentrations of particles at the beginning and end of some experiments.'

Minor Comment 6: 'Table 2 on Page 12. One modal size distribution is given for K-feldspar particles in Table 2. But a two-modal size distribution is shown in Figure S4. Need to explain why the fine mode is neglected.'

The values for both the fine and coarse mode K-feldspar particles are given in Table 2 – Table 2 caption - 'The two values listed between the square brackets are the parameters for the two lognormal modes of aerosol present.' This table has now been moved to Supplementary Table 1, and includes updated, more detailed, lognormal fits to the aerosol present in chamber experiments.

Minor Comment 7: 'Lines 10 - 11 on Page 13. "INPs with four different soluble fractions" could also be given earlier in the model description (see minor comment 3).'

The list of INP soluble mass fraction used in parcel simulations is now given on Page 4.

Changes to the manuscript, Page 5 Lines 4 – 6 - 'In the simulations of chamber experiments K-feldspar particles are given a 1% soluble mass fraction made up of ammonium sulphate. This results in K-feldspar particles having a kappa value of 0.0061 in chamber simulations. In simulations of an adiabatic parcel, the soluble mass fraction of INPs is either 0%, 1%, 25% or 50%.'

Minor Comment 8: 'Lines 29 – 30 on Page 19. "the INPs act as giant CCN" is not as effective as ice formation and subsequent vapor-to-ice deposition for INPs in competition with themselves.'

Although the vapour pressure over an ice surface is less than over a liquid surface, the vapour sink to the INPs acting as CCN is greater than to the INPs which have frozen and are subsequently growing by vapour-to-ice deposition. This is because all INPs act as a vapour sink due to their CCN properties, but only a fraction of INPs nucleate ice and create a vapour sink through vapour-to-ice deposition.

Minor Comment 9: 'Lines 4 -5 on Page 20. This statement is confusing.

"This indicates that further investigation is required into the criteria for heterogeneous freezing as here we have shown it can be the difference between ice formation and no ice formation."

Changes to the manuscript, Page 18, Lines 25 – 28 - 'This indicates the need for further investigation into the criteria for heterogeneous freezing as we have shown that the fraction of frozen ice nuclei in simulations where CCN are present, varies significantly depending on the freezing criteria applied in the model, in some cases it can be the difference between ice formation and no ice formation.'

Minor Comment 10: 'Figure S1. Why are the curves for frozen fraction not monotonic as a function of INP diameter (in units of nm)?'

A more detailed explanation of Supplementary Fig. S1 is now included in the Supplementary material.

Changes to the manuscript, Supplementary Material Page 1, Lines 2 – 25, Page 2, Lines 1 - 3 - 'Three bin structures are compared in Supplementary Fig. S1 which reveals sensitivities to the bin structure used. The full moving structure allows particles in each size bin grow to their exact size. Particles are not moved between size bins. Instead the size of the bin changes as particles grow (or shrink). This structure does not allow processes such as collision coalescence or particle nucleation as the appropriate size bin for new particles may not exist as bin sizes continually change. However the full moving structure provides the most accurate representation of microphysical processes as particles retain their exact size. The other two bin structures in Supplementary Fig. S1 move particles between fixed size bins, and therefore do not allow particles to retain the exact sizes they grow (or shrink) to. In the quasistationary and moving centre bin structures particles grow to their exact size in one time step and are then fitted back onto grid of fixed bin sizes. The method in which particles are fitted back onto a grid varies between the two structures (see Jacoson (1999) for more details). Using a fixed grid allows collision coalescence and particle nucleation to be represented as the appropriate size bin for new particles always exists. However particles do not retain their exact size and are instead put into a size bin closest to their exact size or their volume is averaged between two adjacent bins. This can result in irregularities and numerical diffusion in results. For this reason and the fact that we do not consider particle nucleation or collision coalescence in our study we use the full moving structure in order to provide the best representation of particles in our simulations.

Supplementary Fig. S1 reveals a non-monotonic relationship between INP diameter and frozen fraction. Focusing on the 'full moving ice' line (solid black line), the frozen fraction initially increases with particle diameter. This is because the concentration of ice active sites increases with surface area, i.e. the potential for a particle to nucleate ice increases. At the same time the fraction of activated drops decreases with increasing particle size. This is because larger particles are not able to activate due to kinetic limitations to growth. As particle size increases between approx. 800 nm to 1000 nm the number of ice crystals decreases. Here the ability of a particle to nucleate ice is not limited by the number of active sites presents as the particles have a sufficiently large surface area, but are limited by the mass of water condensed onto their

[Figure]

surface. These particles are not able to achieve the threshold mass of water required for freezing to proceed. As particle size increases beyond 1000 nm, the fraction of frozen particle increases. Here particles are sufficiently large to be able to achieve the threshold mass of water required for freezing, and thus the frozen fraction increases with particle diameter. At larger particle sizes the mass of water required for freezing is less than that required to activate into a cloud drop, therefore the fraction of activated drops decreases with particle diameter.'

Minor Comment 11: 'Figure S16. Is the unit for Dp micron rather than nm? Also the caption is not clear: what are the chamber conditions (e.g. soluble mass fraction?), what is the difference between (a) and (b)?'

The units for Dp should be micron, this has been changed.

The difference between (a) and (b) is temperature, -25oC for chamber conditions and -7oC for parcel simulations. These values have been added to the figure caption.

Minor Comment 12: 'Typos. "approached" on Line 9, Page 4. "location long" on Line 5, Page 8. "where" Line 8, Page 13. "reach" on Line 3, Page 20. "its" on Line 6, Page 20.'

All typos mentioned above have been corrected.

References:

Chuang, P. Y., Charlson, R. J., and Seinfeld, J. H. (1997). Kinetic limitations on droplet formation in clouds. Nature, 390:594–596.

Frey, W., Hu, D., Dorsey, J., Alfarra, M. R., Pajunoja, A., Virtanen, A., Connolly, P., and McFiggans, G.: The efficiency of secondary organic aerosol particles to act as ice nucleating particles at mixed-phase cloud conditions, Atmos. Chem. Phys. Discuss., https://doi.org/10.5194/acp-2017-1223, in review, 2018